# A Novel Concept of the “Standard Human” in the Assessment of Individual Total Heart Size: Lessons from Non-Contrast-Enhanced Cardiac CT Examinations

**DOI:** 10.3390/diagnostics15121502

**Published:** 2025-06-13

**Authors:** Maciej Sosnowski, Zofia Parma, Marcin Syzdół, Grzegorz Brożek, Jan Harpula, Michał Tendera, Wojciech Wojakowski

**Affiliations:** 1Unit of Noninvasive Cardiovascular Diagnostics, Faculty of Medical Sciences in Katowice, Medical University of Silesia, Poniatowskiego 15 Street, 40-055 Katowice, Polandmsyzdol@sum.edu.pl (M.S.); 2Unit of Diagnostic Imaging, Upper Silesian Medical Center, Ziolowa 45 Street, 40-635 Katowice, Poland; 3Department of Epidemiology, Faculty of Medical Sciences in Katowice, Medical University of Silesia, Poniatowskiego 15 Street, 40-055 Katowice, Poland; gbrozek@sum.edu.pl; 4Department of Cardiology and Structural Heart Diseases, Faculty of Medical Sciences in Katowice, Medical University of Silesia, Ziolowa 45 Street, 40-635 Katowice, Poland; jan.harpula@sum.edu.pl (J.H.); michal.tendera@sum.edu.pl (M.T.); wwojakowski@sum.edu.pl (W.W.)

**Keywords:** coronary artery calcium, standard human, total heart volume

## Abstract

**Background:** This single-center retrospective observational study reviewed data from 2305 persons examined for coronary artery calcium (CAC) with non-contrast-enhanced cardiac CT. Other cardiac structures, including chamber volumes, were evaluated besides the CAC scoring. We proposed a novel body size indexing measure that may outperform common indices for quantifying total heart volume (THV). **Methods:** This index is the sum of height and the difference between height (unitless) and body surface area (unitless), [h+(h-BSA)], and if the (h-BSA) equals “zero”, it is a feature of the “standard human”. **Results:** We found that, in subjects with a low cardiovascular (CV) risk, the THV normalized for the novel index was simply a function of BW gain, being the highest in obese. If high-CV-risk features (hypertension, diabetes) were present, the measured THV was larger than expected for BW gain, exceeding values observed in low-CV-risk ones. Differences were found to be sex-independent in all BMI categories. **Conclusions:** Common BSA correction hides these differences and makes the prognostication of CV risk error-introducing. The indexation we proposed might help distinguish the effects of body weight gain from the ones resulting from the presence of certain cardiovascular diseases.

## 1. Introduction

For centuries, heart enlargement has been interpreted as a sign of heart damage caused by various cardiac and non-cardiac diseases. Increased heart size was believed to be compelling evidence of cardiac incompetence. Still, the absence of heart enlargement upon examination does not rule out disease. The apparent normal heart size may occur both in the early stages of the disease or in its mild form, when complications have not yet occurred [1,2,3,4,5].

The reason the averaged absolute and normal range of values is not necessarily effective in assessing heart size in an individual patient is inter-individual variability. It is expected that cardiac dimensions and volumes will vary depending on the individual’s sex and height (h), as well as weight (w) and overall body size, represented by the body surface area (BSA). To reduce the influence of these inter-individual differences, researchers have been looking for the most appropriate method of adjusting measurements of cardiac dimensions and volumes, such as the size of cardiac chambers, their mass and/or volume, and dimensions of the large thoracic vessels, by different indexations [6,7]. For example, the volumetric parameters of the ventricles and atria are often presented as values adjusted to BSA. Height to the power of 2.7 is the advised denominator for indexing left ventricular mass (LVM). Indexation for squared height (h^2^) and height itself is considered the most appropriate for the assessment of the left atrial volume (LAV) [8,9] and aorta size [10], respectively. Interestingly, the body mass index (BMI), the commonly accepted indicator of nutritional status, is rarely used as a normalization factor for the size of the heart and vessels [11].

Given the increasing prevalence of obesity, normalization of anatomical parameters in overweight individuals is of utmost clinical importance, especially in the co-existence of cardiovascular disease, since its natural course may result in heart enlargement. So far, different denominators have been established based on well-documented prognostic studies, with predefined clinical events as their end points [7]. However, separating the effects of obesity alone from cardiovascular disease appears to be an insurmountable challenge [12].

Coronary artery calcium (CAC) scoring has been introduced as a useful tool for the early detection of subclinical atherosclerosis in vivo over the last four decades and has been recommended as a valuable coronary risk modifier [13,14,15,16]. Without exposing the patient to a contrast agent and with minimal exposure to X-ray radiation, non-contrast-enhanced cardiac computed tomography (NCE-CCT) provides information on the presence and, if present, the burden of coronary atherosclerosis, but also enables the assessment of anatomical structures, including precise measurements of dimensions and volumes. Thanks to that, assessing the size of the whole heart, its chambers (ventricles, atria), large thoracic arteries and veins, and epicardial fatty tissue is possible [17,18,19]. According to vague calculations, approximately 33 million patients are estimated to be eligible for CAC examination in the United States alone [20]. A considerable proportion of these patients, especially asymptomatic ones, with no coronary calcium detected, could avoid non-invasive and/or invasive coronary arteriography, as well as unique potentially nephrotoxic contrasting agents and possible invasive procedure-related complications. Since the number of patients undergoing CAC testing is constantly growing, it becomes possible to determine additional indicators, among which heart size seems to be of particular importance, in a large population of people without any additional biological and procedural costs.

Based on the data from NCE-CCT studies performed in our center, we designed a single-center, retrospective, observational study. The acknowledged associations between heart size and body weight and height prompted us to re-evaluate these proportions, giving priority to those body size parameters that are more stable in adults. We assumed that, in adults, height has been a relatively stable parameter of body size over decades [21], and one that only decreases slightly and to a relatively small extent in aging people. Meanwhile, body weight, determined primarily by the type of body structure, is much more variable, depending mainly on nutritional habits and physical activity. Therefore, the priority was to normalize cardiac size, hence the total heart volume (THV, ml), to height and then to normalize it to relative body weight exceeding due weight values attributable to height.

We hypothesized that the “standard human” body weight to height is expressed by the value of body surface area, which numerically (unitless) equals height. Accordingly, when the difference between height (e.g., 1.73 [m], actually unitless) and the BSA value (e.g., 1.73 [m^2^], actually unitless) is equal to “zero”, it is an indicator of the “standard human”. Any deviation from this value determines the share of body mass regardless of height.

## 2. Materials and Methods

### 2.1. Study Population

Based on the data from non-contrast-enhanced cardiac computed tomography (NCE-CCT) studies performed in our center, we designed a single-center, retrospective, observational study. We analyzed a sample of 2305 people who had NCE-CCT performed to determine the coronary artery calcium score (CACS) between 15 August 2008 and 30 September 2023. Of the original cohort, data from before 3 June 2020 (1136 cases) were analyzed de novo, while data from 3 June 2020 onward, when the protocol was extended to include the assessment of the size of the heart chambers (1169 cases), were included under the patient’s records (Table 1). Patients included in the study were referred to our center for CACS assessment by their cardiologist. They were either asymptomatic individuals with coronary artery disease risk factors or with a positive family history, or symptomatic patients presenting with atypical symptoms, such as atypical or non-anginal chest pain, exertional dyspnea, or the presence of unexplained arrhythmia. We also included symptomatic individuals referred for contrast-enhanced coronary CT angiography, in whom a standardized non-contrast pre-test revealed advanced calcified lesions. The severity of calcification significantly limited the interpretive value of contrast-enhanced imaging, making the CAC score effectively the final diagnostic tool in these patients.

### 2.2. Cardiovascular Risk Factors’ Evaluation Se

For each subject, age (y), sex (f/m), height (h, m), weight (w, kg), body mass index (BMI, kg/m^2^), smoking habits, systemic arterial blood pressure, lipid levels, and diabetic state were noted according to a simple risk assessment protocol. Normal weight (BMI 17.8–24.9 kg/m^2^), overweight (BMI 25–29.9), and obese (BMI ≥ 30) categories were used. Smoking habit was categorized as ever (currently or anytime in the past) and never smoking. Systemic arterial hypertension (SAH) was recognized in subjects diagnosed as hypertensive as yet, those who were treated with hypotensive drugs, or in whom blood pressure measured before CT examination was equal to or more than 180/100 mmHg or required the administration of BP-lowering drugs. Dyslipidemia (HL) was diagnosed in subjects taking lipid-lowering medications or who had a documented plasma total cholesterol (TC) of 200 mg/dL or more, or triglyceride (TG) of 150 mg/dL or more, or both. Diagnosed type 2 diabetes mellitus (T2DM) was recognized only in subjects treated with insulin and/or oral agents. Patients with abnormal oral glucose tests or fasting glucose elevation were considered non-diabetic unless treated. There were no patients with type 1 diabetes.

### 2.3. Coronary Artery Calcium Determination

Coronary artery calcium (CAC) examinations were performed without the use of any contrast agent (non-contrast-enhanced, NCE-CCT) using a 64-row MDCT (Aquilion, Toshiba, Tokyo, Japan) or 2 × 192-row (SOMATOM Force Dual Source, Siemens Healthcare GmbH, Erlangen, Germany) with a predefined heart-rate-adjusted prospective scanning in the mid-diastolic phase (MD). Multi-planar (2D and 3D) offline reconstructions of the images were performed on Vitrea 2 workstations (software version 3.9.0.0, Vital Images, Hopkins, MN, USA) or Syngo.via (version VB30A_HF06). The coronary calcification burden was assessed manually or semi-automatically following the method of Agatston et al. [13]. A commercially available standardized reporting format for CACS was used. The total CACS was chosen for further analysis (Table 2).

### 2.4. Heart Size Evaluation

To assess the heart size, we used commercially available software on the Vitrea 2 workstation with a “general” module and an “organ” function, enabling semi-automatic determination of the volume of a designated anatomical organ or its part using a 3D object detection technique (based on self-adjusted automatic Hounsfield units’ window). Manual correction was required in almost all cases. The mid-diastolic (MD) fatty-less total heart volume (THV) was defined as the sum of the volumes of the left and right atria (LAV + RAV) and ventricles with the interventricular septum (biventricular volume, BIV). Ventricular volumes included the outflow tracts and ventricular mass. Volumes of the left and right atria included appendages (Figure 1). Data were also presented as a sum of atrial volumes (bi-atrial volume, BAV) Appendix A. The fatty tissues of the heart, epicardium, aortic root, and pulmonary artery were excluded.

### 2.5. Inter- and Intra-Observer Variability

All measurements were performed by one experienced investigator (MS, >7 K manually corrected measurements, which provided >11.5 K individual parameters of the cardiac chambers and THV). The intra-observer reproducibility of the measurements was examined in 125 subjects, randomly selected by the primary investigator, on calculations made from 6 weeks to 2 years apart from the first evaluation. The inter-observer reproducibility was assessed in 45 subjects by two independent investigators (MS, JH) using the same criteria. The results are presented in Table 3.

### 2.6. Physiological Meaning of h-BSA

The difference h-BSA is, essentially, the difference between two numbers. The first is the numerical (without measurement units) value of height; the second is the numerical (without measurement units) value of the calculated body surface area. To explain its biological meaning, we examined its relationship with the body mass index. We found that the height–BSA difference was inversely related to BMI (Figure 2).

### 2.7. Study Subgroups

First, we separated a reference group that included 353 non-overweight/non-obese people (BMI < 25 kg/m^2^) with low CV risk defined as a CACS of “zero” or less than 10 Agatston units (AU), absence of t2DM and SAH, with no other risk factor or no more than one risk factor of the two, i.e., cigarette smoking (current or former) or dyslipidemia (HL, high TC with normal or high TG). In these subjects, the post hoc review of NCE-CCT images confirmed the absence of any obvious cardiac anatomical abnormalities. The remaining patients were assigned to subgroups depending on the presence of risk factors (CACS ≥ 10 AU or, irrespective of CACS, the presence of SAH or t2DM, irrespective of the presence of other risk factors, and simultaneous smoking and HL irrespective of the presence or absence of CACS, SAH, t2DM or obvious anatomical abnormalities) and nutritional status according to the BMI category. The characteristics of the subjects are presented in Table 2.

### 2.8. Relationship Between Total Heart Volume and Height in the Reference Group

Height and body size area have been chosen among the factors determining the total heart volume. The empirical relationship was established as a result of multiple regression analysis, the regressors of which were “h” and the difference between “h” and BSA, disregarding units, and the dependent factor was THV, as follows (Equation (1)):THV (mL) = 456 × h − 221 × (h-BSA) − 335 (±142) (n = 353. R = 0.56. F = 80.2. *p* < 0.001)(1)

### 2.9. The “Standard Human”

“Standard human” (sh) was defined as an individual whose numerical value of height (unitless) was equal to the numerical value of BSA (unitless); i.e., the difference between these two numbers equaled “zero” (h-BSA = 0), and the expected THV value was dependent on height only, according to the following Equation (2) (case A in Table 4):THV (“standard human”) = 456 × h − 335 (i.e., for h = 1.73 m, w = 62.28 kg (~62.3 kg))(2)

The difference between the reference value (THVsh) and the expected value specific to height and h-BSA is closely related to body weight. In a person with h = 1.73 and BSA = 1.53 and a weight of 48.7 kg, the expected THV equals THVsh − 221 × (0.2) (case B in Table 4). In a person with height = 1.73 and BSA = 1.93 and a weight of 77.5 kg, the expected THV equals THVsh − 221 × (−0.20) (case C). Height was 1.73 m in all. With increasing BMI, the expected THV increased alongside (case D). The upper prediction limit (UL) is a reference value designated to detect THV greater in relation to the body mass increase only.

**Table 4 diagnostics-15-01502-t004:** Examples of calculation of the expected total heart volume depending on height and h-BSA difference.

Case	Height [m]	Weight [kg]	BMI [kg/m^2^]	BSA	h-BSA	Expected THV [95% Prediction Interval]
A	1.73	62.3	20.8	1.73	0	453 [312–595]
B		48.7	16.3	1.53	0.2	409 [264–555]
C		77.5	25.9	1.93	−0.2	498 [356–639]
D		89.0	29.8	2.07	−0.34	529 [387–671]

### 2.10. Individual-Oriented Approach

The upper normal limit of the expected total heart volume (THV_UL_, ml) was calculated according to the empirical equation in the reference subgroup (Equation (3)), as follows:THV_UL_ = 456 × h − 221 × (h-BSA) − (335 − 142), thus THV_UL_ = 456 × h − 221 × (h-BSA) − 193 (3)

In people for whom the mean difference between measured THV_m_ and THV_UL_ was lower or equal to zero, heart size was assumed to be explainable by height and weight. Any excess in THV other than that in relation to body size, as indicated by the difference THV_m_ − THV_UL_ > 0, was considered as a possible influence of a co-existing pathological state or process. This allows the impact of disease factors to be considered along with absolute (mL) and relative (%) quantification. Additionally, in each case, a partial THV value could be provided by which the value indexed to the BSA (THV_H_-TVH_BSA_) should be corrected (Figure 3). Reverse correction is also possible. The BSA was calculated after the Mosteller equation, as [(h × w)/3600]^0.5^ [22].

### 2.11. Statistical Analysis

Quantitative data are presented as mean ± 1 standard deviation or median and interquartile range, depending on the distribution. Qualitative data are presented as numbers or proportions. The normality of distribution is based on the Shapiro–Wilk test. Student’s *t*-test was used to compare normally distributed data from two groups and the Kruskal–Wallis test for their non-parametric comparisons. ANOVA/MANOVA was used for more than 2 groups’ comparisons. The chi-square test was used to quantify distributions. Multivariate logistic regression analysis was used to determine predictors or co-factors of THV. For reproducibility analysis, mean errors, intra-class correlation coefficients, and limits of agreement were calculated. Pearson correlation was used for normally distributed quantitative data. Coefficients of determination were calculated in addition. A statistical package was used to perform the calculations (Statistica 8.0, StatSoft Inc., Tulsa, OK, USA). Statistical significance was based on a *p*-value < 0.05.

### 2.12. Ethics

This study was conducted in concordance with the Helsinki Declaration. The present investigation gained a priori ethical clearance from the Silesian Medical University Ethics Committee, according to its retrospective design and the anonymity of the individuals examined. Nevertheless, in principle, patients gave informed consent to undergo X-ray examinations in our hospital center for diagnostic reasons. They signed an agreement to access their anonymized data for future retrospective analysis.

## 3. Results

### 3.1. Total Heart Volume in the Reference and Other Subgroups

Age was similar irrespective of sex in the reference group (Table 2). Age was similar in subjects with low risk, irrespective of BMI. The mean age of females and males with high risk was higher than those with low risk (65.9 ± 10.0 vs. 60.0 ± 10.4, *p* < 0.001). Males were younger than females in other subgroups than the reference (57.5 ± 11.4 vs. 61.1 ± 9.5, *p* < 0.001). Height was similar in all comparable subgroups, while body weight was within the range of BMI classes (Table 2). Thus, the mean H-BSA difference was greater in men. In the reference subgroup with low CV risk and the subgroup with normal BMI and higher CV risk, these parameters were similar, with lower means in women than in men (Table 5). Overall, the THV was smaller in females (449 ± 99 mL) by approximately 150 mL (~25%) as compared with males (594 ± 141 mL, *p* < 0.001). The differences in subgroups are shown in Table 6.

After indexing the measured THV to BSA or height, the sex-related differences remained. In women (247 ± 49 mL/m^2^ and 278 ± 60 mL/m), they were lower compared with men (287 ± 60 and 339 ± 77, resp.) by approximately 40 mL/m^2^ for the THV/BSA and 50 mL/m for the TVH/h. Significant differences between subgroups are shown in Table 6.

Detailed data regarding measured LA, RA, BAV, and BIV are presented in Appendix A. These partial volumes were also significantly smaller in women than in men. Indexation for height showed a clinically marginal but statistically significant lower right atrial volume (~2 mL/m) in women. A greater difference was found for BIV (~36 mL/h) and THV (~41 mL/h) with values higher in men (Appendix A). The partial contribution of cardiac chambers was ~1/3rd for bi-atrial and ~2/3rd for bi-ventricular volumes in the reference subgroup.

In the “standard human”, the THV was lower in women by approximately 60 mL, with a relative difference of 15%. In females with low CV risk, the average measured and expected THV values were close to the “standard human” condition. In males with low CV risk, the average measured, expected, and “standard” THV differed by approximately 20–40 mL (Table 7). This was in full accordance with the H-BSA difference.

### 3.2. THV Relations with Body Mass Index and CV Risk

In females with normal BMI and low CV risk, the standard, measured, and expected values of THV were nearly identical. In normal females with high CV risk, measured THV (THVm) was on average 25–30 mL greater, actually, only marginally higher if compared with the expected or “standard” values (431, 408, and 402 mL, respectively). In overweight females with low CV risk, measured and expected THV (THVex) values were marginally higher than the THV “standard” values (i.e., 427, 443, and 404 mL, respectively). In obese women with low CV risk, these differences were significantly greater than the standard values (i.e., 452, 484, and 402 mL).

In overweight females with high CV risk, the measured and expected THV values were marginally higher than the “standard human” values (i.e., 427, 443, and 404 mL, respectively). In obese women with low CV risk, these differences were significantly greater than the “standard” values (i.e., 452, 484, and 402 mL, respectively) (Figure 4A,B).

In overweight low-risk males, the raw and expected values were similar but far from the “standard” values (525, 523, and 463 mL, respectively). In overweight women, these differences were clear (452, 438, and 399 mL, respectively), while in obese women, the differences were greatest (514, 483, and 398 mL, respectively). Similar comparisons in the high-risk males with obesity showed that the measured THV was higher than expected by 80–120 mL on average and by 120–190 mL on average as compared with the “standard THV” (Figure 4A,B). Additionally, all comparisons between individuals with low risk and high risk, showed that THVm was significantly greater than THVex in the last groups (Table 7).

### 3.3. Prevalence of Abnormal THV

In patients with low CV risk, the proportion of abnormally high THV was low, according to the criteria used, irrespective of the BMI class (Table 8).

The total heart volume lying above the empirical upper normal limit (THV_UL_ > 0), according to the established height and height–BSA difference, was found in 15.4% of the entire group (Table 8). THV_UL_ > 0 was more prevalent in males (20.3%) than in females (6.3%) and in high-risk compared with low-risk patients (24.8% vs. 3.5%) (*p* < 0.001, both). In patients with systemic arterial hypertension or those with a CAC score ≥ 10 Agatston units, similar statistically significant disproportions were noticed (16.6% vs. 11.8% and 17.8% vs. 10.9%, respectively). In patients with type 2 diabetes mellitus, there were no differences in means.

In any case, our data indicated that, on average, the indexation for the BSA provided calculated THV to be lower in comparison with the THV indexed for height in a whole sample, as well as in specific conditions (Table 9). In patients with SAH or T2DM, regardless of smoking or HL, an increase in cardiac size was greater than that resulting from increased body weight.

Analyzing the relationships of THV normalized by height or body surface area and the difference between the two indexed THV values and the expected upper limits of normal in the entire sample, the trend for increasing THV with sex and BMI was abolished while using the indexation for BSA in both quality categories. Still, these values were similarly higher in the group of patients with THV above the upper-limit category (Table 10). Comparing the results in the context of the presence of some factor-oriented modifiers such as sex and risk factors (SAH, t2DM, or CAC), the differences of statistical significance were observed every time for THV expected on the basis of the equation including the “h-BSA” difference (Table 10).

## 4. Discussion

### 4.1. Main Result

The most original achievement of our study is the “standard human” concept that introduces a novel denominator for the indexation of THV to personalize the result. The “standard human” is a statistical subject that has equal unitless values (numbers) for height (i.e., 1.73) and BSA (i.e., 1.73). The arithmetic difference between these two numbers (called h-BSA) above “zero”, normally seen in the vast majority of individuals, indicates that body weight differs actually from that related solely to height for many reasons, like sex difference (at the same height, males weigh more), physical activity habits (more active people usually have a bigger heart), or fat tissue accumulation (overweight and obesity), among others. Many pathological conditions or their sequelae are associated with an increase in the whole heart size. The finding of THV above the relatively wide upper limit of normality in a non-obese person that cannot be explained by physical activity may indicate an underlying disease, even if the absolute values appear to lie within the normal range guided by recommendations. We found that in high-risk patients, the measured THV was larger than expected, remarkably exceeding values observed in those with low risk. These differences have been found to be sex-independent in any BMI category. Our observation is quite new and has never been communicated previously.

The important asset of the “standard human” heart size is that the proposed equation, knowing the height and weight of individuals or the sample averages, allows for the re-calculation of normative data obtained by using various methods evaluating cardiac size and/or specific chamber size wherever these parameters were used for indexation. Our results confirmed that the indexation of the whole heart (as well as its constituents) for BSA introduces errors in overweight/obese people, which can actually be estimated and overcome by using the “standard human” concept we proposed. Since the presented empirical equations referred to a specific population sample, the average THV for a different sample should be adjusted accordingly.

### 4.2. Methodical Issues

The results of our study have brought important information that requires explanation in some aspects, like the phase of the cardiac cycle in THV imaging and chambers selected during CT acquisition, cardiac CT imaging methods (with or without contrast enhancement), the purpose of the study, and patient selection, among others.

First of all, and most importantly, is the phase of acquisition of the cardiac scans. All the images we analyzed were acquired in mid-diastole, which is typical for NCE-CT heart scans. No normalized data from sample or population-based studies for this phase of the heart cycle have been established so far.

The reason for the lack of these data is related to commonly used methods of cardiac assessment. Typically, normative data concerning heart volumes come from analyzing end-systolic (ES) and end-diastolic (ED) phases [8,9]. While normative volumes for ventricles are calculated during diastole, values concerning the LAV and RAV usually come for the end-systolic phase. Relying on current data, it is not easy to calculate and determine the THV since the ventricles and atria sizes are recommended to be calculated at total opposite phases of the heart cycle. [8,9].

Our study is the largest analysis of raw and indexed data of total THV determined manually from a mid-diastolic NCE-CT scan, performed before the era of artificial intelligence [23]. In the study of Walker et al. [24], the LV and LA volumes at the MD phase (fixed prospectively at 74% of a cardiac cycle) accurately estimated their maximum values (ED and ES, respectively), while evaluated from contrast-enhanced CCTA images. Normal values for the MD period were proposed, derived from a group of 101 normotensive individuals (35% women) without diabetes or heart abnormalities. The average BSA was 1.94, indicating that overweight and obese people were included in their control group (BMI not given). Additionally, any positive CACS was allowed. Normal MD-LVV and MD-LVM values were on average 128 mL and 116 g for all subjects. Our data on BVV are hardly related to the results of the discussed study, as distinguishing between LV, RV, and LVM was a priori impossible [25]. In the abovementioned study, the overall mean MD-LAV (without appendage) was 68 mL (95% CI 38–99), 64 mL (95% CI 36–92) in women and 71 mL (95% CI 39–102) in men. In our reference (353 people), the mean MD-LAV (with appendage) reached 67 mL (95% CI 52–84), 65 mL (95% CI 51–82) in women and 74 mL (95% CI 61–94 mL) in men. In addition, we optimized the diastolic phase range for cardiac MD volume determination using the adjustment for the HR during a CT examination, which varied between 60% and 80% in subjects with faster or slower HR, respectively.

In a validation study by Mao et al. [26], the reference values for four-chamber volumes and LV mass were determined automatically in the MD phase in 131 normotensive and non-diabetic patients undergoing CE-CCTA. The sample included 59 men and 72 women with a “zero” CACS and normal left ventricle function. The exclusion criteria did not include increased BMI, among others. The reference THV was, on average, 458 mL in females and 569 mL in males. However, since 30% were obese, the validity of the reference may be questioned. In our study, we adopted more stringent criteria; i.e., only people with normal BMI were included in the reference subgroup. The difference was similar regarding sex; meanwhile, the actual volumes were smaller (405 and 507 mL). Additionally, data from the entire study population, regardless of BMI, showed mean THV values of 420 mL in women and 520 mL in men.

Our results, which differed from contrast-enhanced averages by approximately 40–50 mL, are well above the estimation error. What is also important is that, in the study of Mao et al., the average HR was around 58 bpm (63 out of 131 patients were given beta blockers), which is associated with the heart working at a relatively larger preload. In our population, HR at CT acquisition was 68 bpm on average. The difference between cycle durations at these two HRs (1.034 s–0.882 s) reaches 150 ms. This period is long enough to enable a change in the volume of the ventricles, although the quantity of the volume enlargement is not specified. This relation is more evident in men with an average volume decrease of 4% to 5% of for every HR increase of 5 beats/min [27]. Finally, we know that any data comparison obtained by different methods (with contrast enhancement or without it) is subject to method-related inequalities.

In the Copenhagen General Population Study, 597 normal-weight people were examined to establish the range of heart volume values in the MD phase of CE-CCTA images [28]. In 381 females and 188 males, the mean THV reached 381 ± 57 and 492 ± 88 mL, respectively. These values differ from those in our NCE-CCT study by 15–20 mL, which is well within the estimation bias range. After indexation for BSA, the THV was 252 ± 42 and 223 ± 29 mL/m^2^ in normal-weight low-risk males and females in the cited study, respectively [28]. We observed values that differed by approximately 20 mL/m^2^. After indexation for height (recalculated from data [28]), the average THV was 272 and 228 mL/m, respectively. In our study, the average THV indexed for height reached 290 and 249 mL/m. The present comparison may support the equality of THV indexation to height or BSA in non-overweight people.

Juneau et al. [29] proposed normal reference ranges for MD-LV volume and mass in a large cohort of 2647 symptomatic patients, irrespective of their BMI. Among patients who had constituted the “normal” group, 54% were male, 15% were smokers, 47% had hypertension, 14% had diabetes, and 48% had dyslipidemia. Additionally, 12% had an obstructive coronary lesion. In an attempt to explain their approach, the authors wrote, “To overcome the potential limitations of sample size, we included patients with diabetes, hypertension, and obstructive CAD”. In contrast with the findings of Fuchs et al. [28], the recalculated difference between height and BSA (h-BSA) reached −0.343 in males and −0.227 in females. Accordingly, the differences between MD-LV/h (a case of “standard human” with h-BSA = 0) and MD-LV/BSA were around 8 mL/m in females and 12.5 mL/m in males. These may be non-negligible constraints on the application of the proposed normality ranges.

Other observations come from the study of Massalha et al. [30], who aimed to establish reference values for the MD-RV volume from the CE-CCTA images in 1542 patients. Among all participants, only 837 (54%) were normotensive and non-diabetic, irrespective of BMI. The recalculated h-BSA (from the presented data) reached −0.34 in males and −0.217 in females, quite similar to that of the former study [29], as they come from the same center. Again, the differences between MD-RV/h (a case of “standard human” with h-BSA = 0) and MD-RV/BSA were around 9 mL/m in females and 12.3 mL/m in males. Taking into account that the investigators used data from CE-CCTA, our measurements for the BIV cannot be compared. Meanwhile, the sum of differences for MD-LV and MD-RV ranges around 17 mL/m and 35 mL/m in females and males, respectively (without LV mass). A reverse calculation indicated volume differences of ~27 mL in females and 63 mL in males, which link up with body mass difference from the “standard human”. If we compare this value with the sum of reported measured mid-diastolic LV and RV means (F, 224 mL, and M, 305 mL), differences of 12.5% and 20% in females and males, respectively, show how far the references are from the “standard human”.

In another comparative study, Daniel et al. [31] tried to assess whether the NCE-CCT-derived LV parameters correlated with the MRI-derived LV mass in 60 patients of the MESA study cohort with no overt CV diseases. In their study, the calculated bi-ventricular volume was 237 ± 71 mL. In our work, the average BIV reached 293 mL in the reference subgroup (Appendix A). One of the reasons for this difference is the fact that, in the MESA cohort, a 50% phase of acquisition was used, in contrast with our 60–80% range. Nevertheless, the BIV was found to correlate with LVM. It is obvious that since the ventricular and interventricular muscle fibers constitute the ventricular walls, any increase in LVM results in a significant increase in overall BIV volume.

Budoff et al. provided reference BSA-indexed values for volumes of left and right atria in a group of 203 normotensive, non-diabetic patients (121 M, 82 F) [32]. The end-systolic LAV reached 94 mL and 90 mL, respectively. Reported BMI values and recalculated mean h-BSA values of −0.328 and −0.148 indicate that a significant proportion of overweight individuals had been included in the reference group. Additionally, the difference between LAV/BSA (44 (M) and 51 (F) mL) and LAV/h (53 and 56 mL, resp.) was 9 and 5 mL. Similar differences were found for RA end-systolic volumes. For the “standard human”, the indexed values should be equal, also in case of single chamber volumes, dual-chamber volumes, and the THV.

### 4.3. Patient Selection Special Criteria

In the studies included as the ground for the latest recommendations of numerous cardiac imaging societies, various selection criteria were applied. Our study is among those where CAC score was included as a selection criterion. The ESC 2021 guidelines on the prevention of cardiovascular diseases emphasize the important role of the CAC score [15]. Even though CACS is the primary reason for performing an NCE-CT scan of the heart and chest, it is rarely considered as a parameter that reflects the health of the CV system. In a recognizable paper by Lin et al. [33], age- and sex-specific reference data were given for cardiac chambers’ size, function, and LV mass based on CE-CCTA in 103 middle-aged normotensive, non-obese (BMI < 30) adults without CVD, but with subclinical coronary atherosclerosis (mean CACS of 32 AU). In the Copenhagen General Population Study, the positive CACS (54 AU [IQ 13–140]) or the presence of obstructive CAD, seen in every fourth subject, has not been considered as an exclusion criterion [28]. In the cited study of Budoff et al. [32], CACS ≤ 100 AU or ≤50% stenosis in any coronary artery was allowed. In our study, we included people with CACS of less than 10 AU in the reference group because, as such, they have a documented low risk of cardiovascular events [34]. This means, in a sense, that the reference group, although not referring to a group of purely healthy people, is as close as possible to the sample as any available source in setting cardiac tomography standards. Indeed, our selection criteria were rather rigorous, especially concerning body constitution, compared with the former studies that used various imaging techniques.

New possibilities for determining the size of the whole heart from non-contrast-enhanced CAC scans have been opened, thanks to the potential of artificial intelligence (AI). In a validation study by Jacob et al. [35], an automated, AI-based quantified THV from NNE-CCT was compared with data from contrast-enhanced CCTA in 420 symptomatic patients (43% females) with a mean age of 63 ± 13 years. The average TVH from CE-CCTA was 602 mL, which, due to the mean absolute bias of 7% of NCE-CCT volumes (~42 mL), corresponds to an average THV of 644 mL. A similar value (average THV of 641 mL) was found in our study in 474 overweight and obese men (Table 6). According to clinical recommendations, the majority of our patients had no indications for CE-CCTA. However, the comparison (shown above) allows us to forecast that our overweight/obese, high-risk, large-hearted patients in the NCE-CCT study have the same high risk of cardiovascular events as patients with symptomatic cardiomegaly diagnosed in the CE-CCTA study.

In a study conducted by Naghavi et al., the volumes of all heart chambers plus left ventricular mass were determined by AI in 169 paired cases of the same individuals aged 62 ± 10 years, 53% females, who underwent CT for lung cancer screening [36]. The study showed that the THV could be calculated from the NCE chest CT data. Interestingly, the average AI-THV was approximately 506 mL, which is very close to an average THV of 499 mL in 2305 subjects aged 63 ± 11 years, 65% females in our study.

Recently, very promising information has come from the same team of investigators, based on the AI interpretation of CAC scans [37]. Results of the study, which was performed as an extension of the Multiethnic Study of Atherosclerosis (MESA), indicate that increased THV outperforms the estimation of the risk of heart failure (HF) by NT-pro-BNP concentration and CAC score. In 5750 participants, the average THV was 482.3 ± 108.7 mL. In 5194 persons in whom HF did not develop over 15 years of follow-up, the average THV was 478.1 ± 105.8 mL. In 256 patients who presented with HF over this period, the initial value of the THV was 541.1 ± 132.3. In our study, even greater mean values were observed in high-risk obese patients.

### 4.4. Limitations

A measurement error of 10–20 milliliters may be considered irrelevant, as the difference between compared subgroups reached up to 150 mL. Still, one has to realize that the assessment of various heart structures based on NECCT without contrast enhancement is difficult. The contours of areas corresponding to analogous structures in contrast-enhanced studies are much more blurred, which may make reliable assessment more challenging. This might be of importance in the case where the measured or indexed value was borderline (small THV values/hearts), and we assumed that this group of patients would be less than 5% of the population.

Our results are indeed not comparable with the findings of other researchers, as to the best of our knowledge and in good faith, a manually determined total heart volume has not been described from studies with NCE-CCT yet. Nevertheless, results from a single center and specific ranges of normalcy derived from and relating to the studied population sample cannot be extrapolated and transferred to other populations, including ethnically different or multi-ethnic ones.

Unfortunately, we were unable to separate right and left ventricular volumes and left ventricular mass in NCE-CT. However, novel proposals for LVM delineation of NCE scans and LV hypertrophy assessment have appeared with the use of AI algorithms [38,39]. The AI-derived cardiac chambers and LV mass parameters were shown to be reliably measured from chest CT scans and provided important prognostic information [40].

### 4.5. Perspectives

The idea presented in our work is that a “standard human” can be described by the difference between height and h-BSA that equals zero. In healthy people, any non-zero value might reflect sex differences (woman, man), body type (asthenic, athlete, picnic), and physical activity (inactive, rarely active, regularly active, trained, sportsman). THV far from the “standard human” measured in an athlete should not indicate any disease, but should rather be expected. Normalized individual values for height and height-BSA difference can help to confirm data accuracy or indicate other reasons for heart enlargement.

The rules we propose might help to recalculate data, in which cardiac morphological parameters indexed for body surface area are presented. The correction volumes can be easily recalculated using height. Thus, hopefully, historical data can be re-interpreted. This is of special importance for the evaluation of cardiovascular status in patients with obesity.

An important advantage is that although our data focused on cardiac chambers and total cardiac volume in the mid-diastolic phase, the “standard human” approach can be widely used to quantify the size of the cardiac chambers or the entire heart in any phase, including the commonly used end-diastolic phase for the ventricles and end-systolic phase for the atria. In each such situation, the researcher can accurately answer the question of how the measured ventricle, atrium, or the entire heart fits in the body. As height and BSA have been universally used, their difference might be used to recalculate historical data, for which the outcomes are known, and their value can be assessed and compared with common correction indices.

Thanks to NC-CCT analysis, another hope might be given to numerous patients who have limited acoustic windows for performing echocardiographic imaging, in whom poorly visualized anatomical structures could be visualized [41]. Indeed, non-contrast-enhanced cardiac CT with ECG gating might become the first step in the assessment of the cardiovascular anatomy at a relatively low X-ray exposure risk.

With the latest advancements in artificial intelligence, it has become possible to accurately calculate the heart volumes in hundreds and thousands of individuals who undergo the CACS test. Recent studies provide evidence for this claim [37,40]. However, using the concept of the “standard human” could result in a personalized assessment for each individual.

## 5. Conclusions

A novel concept of the standard human is presented. A simple equation enables us to separate the effects of body weight related to height from relative body weight above that attributable to height. The proposed method might be helpful for the individual interpretation of cardiac size. It is of special importance in patients with body weight excess, in whom the clinical interpretation of cardiac size related to body mass and co-morbidities is challenging.

## Figures and Tables

**Figure 1 diagnostics-15-01502-f001:**
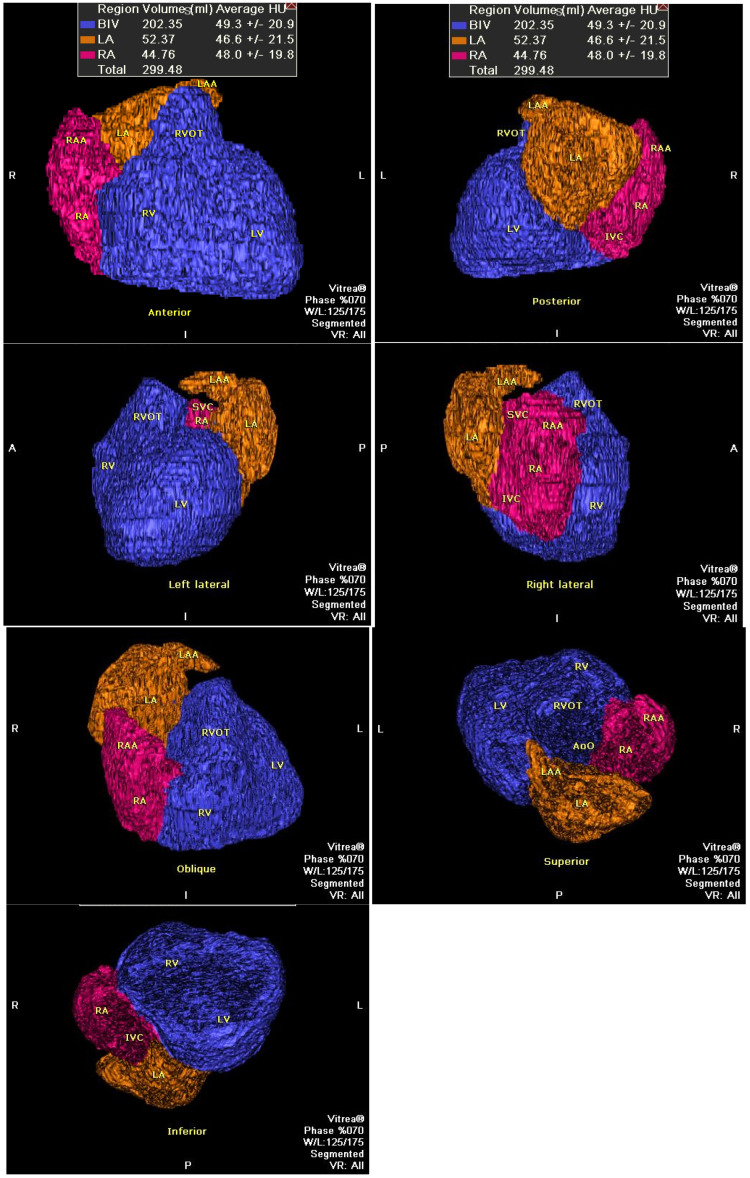
Multiplane projections of the total heart volume. Abbreviations: BIV—bi-ventricular, IVC—inferior cava vein, LA—left atrium, LAA—left atrial appendage, LV—left ventricle, RA—right atrium, RAA—right atrial appendage, SVC—superior cava vein, RVOT—right ventricular outflow tract.

**Figure 2 diagnostics-15-01502-f002:**
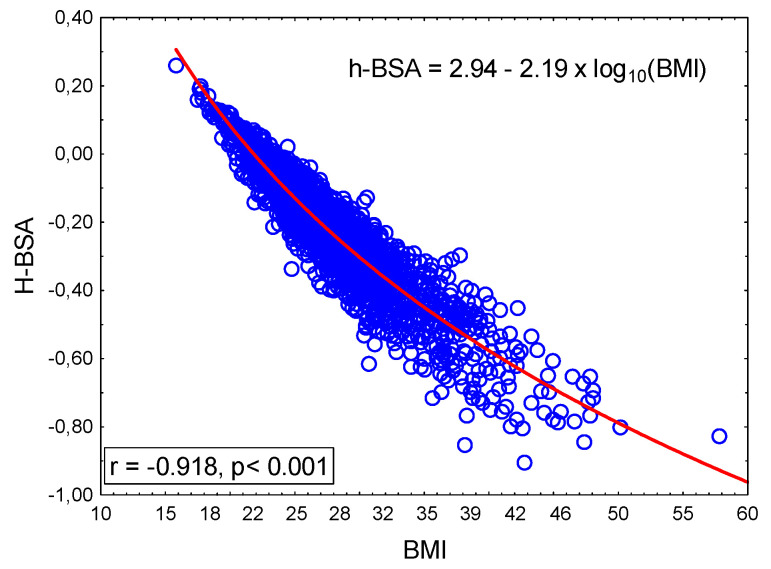
Relationship between BMI and h-BSA.

**Figure 3 diagnostics-15-01502-f003:**
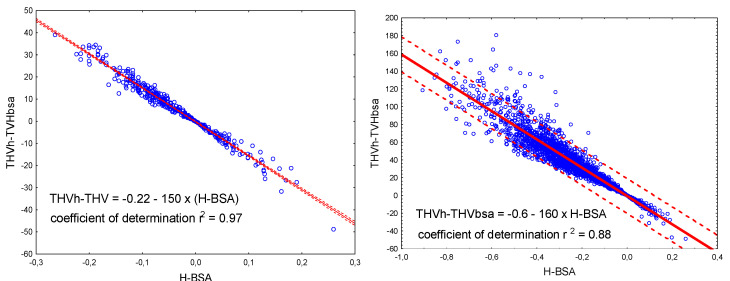
Scatterplots of h-BSA difference versus THVh-THVbsa indexed values in the reference subgroup (**left**) and in the entire group (**right**) with mean (solid line) and prediction limits (dashed line).

**Figure 4 diagnostics-15-01502-f004:**
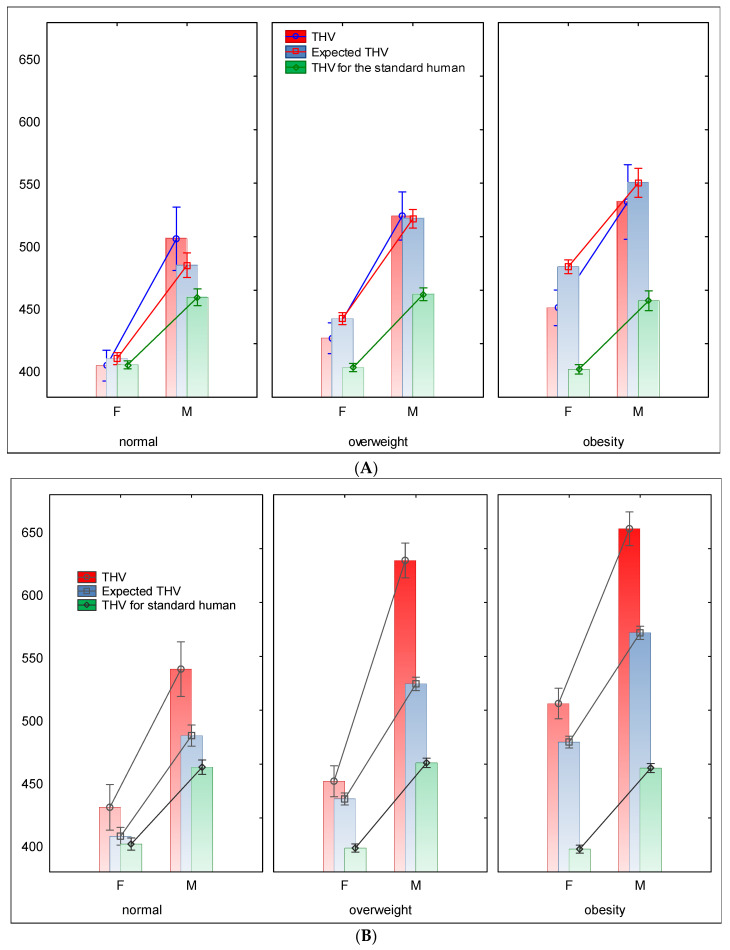
(**A**) Total heart volume depending on sex and BMI in subjects with low CV risk. (**B**) Total heart volume depending on sex and BMI in subjects with high CV risk. Abbreviations: F—female, M—male, THV—total heart volume [mL]. Statistics: MANOVA Wilks lambda = 0.99, F = 2.40, *p* = 0.025, effective hypothesis decomposition. Bars represent the least square means, and vertical bars denote 0.95 confidence intervals. Detailed data are presented in Table 7.

**Table 1 diagnostics-15-01502-t001:** Characteristics of the examined population.

Parameter	Mean/Proportion
Age (years)	63 ± 11
Sex (females, F/males, M)	1503/802
Risk factors (count)	1.7 (0–4)
Smoking	494 (21%)
Systemic arterial hypertension (SAH)	1728 (75%)
High lipids (HL)	1226 (53%)
Type 2 diabetes mellitus (t2DM	458 (20%)
Body mass index (BMI, kg/m^2^)	28.6 ± 4.9
Normal	579 (25%)
Overweight	923 (40%)
Obese	803 (35%)
Coronary artery calcium score (CACS) > 0	1463 (63%)

**Table 2 diagnostics-15-01502-t002:** Anthropomorphic data concerning sex and BMI.

Sex	BMI Category	*n*	Age	BMI	BSA
F	0	418	62 ± 10	22.87 ± 1.72	1.65 ± 0.11
	1	575	66 ± 10	27.44 ± 1.44	1.79 ± 0.11
	2	510	64 ± 10	34.18 ± 3.87	1.99 ± 0.15 ^a^
M	0	161	62 ± 12	23.37 ± 1.72	1.86 ± 0.13
	1	348	62 ± 11	27.59 ± 1.41	2.03 ± 0.14
	2	293	60 ± 11	33.44 ± 3.26	2.22 ± 0.16 ^b^

Abbreviations: F—female, M—male, BMI—body mass index [kg/m^2^], 0—normal weight, 1—overweight, 2—obese, BSA—body surface area [m^2^], statistics: ANOVA ^a^—F(4, 2998) = 7553, *p* < 0.001 (females), ^b^—F(4, 1596) = 386, *p* < 0.001 (males).

**Table 3 diagnostics-15-01502-t003:** Reproducibility of THV.

Parameters	Mean Difference	ICC	LOA	Pearson’s r
THV_0_ vs. THV_1_ (*n* = 45)	−15.6 mL	0.996	−51.0–19.8	0.992
	−3.1%		−9.4–3.2	
THVa vs. THVb (*n* = 125)	−1.7 mL	0.996	−33.2–29.8	0.991
	−0.4%		−6.5–5.6	

Abbreviations (inter-observer (2 readers, 0 vs. 1) and intra-observer (within the reader “0”, a vs. b) concordance: THV—total heart volume, CI—confidence interval, ICC—intraclass correlation coefficient, LOA—limits of agreement.

**Table 5 diagnostics-15-01502-t005:** Mean values of h-BSA difference in females and males according to CV risk and BMI.

Sex	Female			Male		
Risk	Low			Low		
BMI	Normal	Overweight	Obesity	Normal	Overweight	Obesity
h-BSA	−0.022 ± 0.072	−0.177 ± 0.061	−0.372 ± 0.106	−0.116 ± 0.072	−0.274 ± 0.070	−0.427 ± 0.094
Risk	High			High		
BMI	Normal	Overweight	Obesity	Normal	Overweight	Obesity
h-BSA	−0.028 ± 0.068	−0.177 ± 0.061	−0.385 ± 0.125	−0.114 ± 0.074	−0.284 ± 0.070	−0.487 ± 0.119
Significance	p < 0.0001			p < 0.0001		

Abbreviations: BMI—body mass index class; BSA—body surface area; h—height.

**Table 6 diagnostics-15-01502-t006:** Measured and indexed total heart volume (fat-free) in all subgroups.

BMI Class	Risk	Sex	N	THV	THV/BSA	THV/h
0	0	F	285	405.3 ± 73.0	246.2 ± 42.1	249.3 ± 43.1
0	0	M	68	506.9 ± 89.5 ^c^	271.7 ± 40.8 ^c^	290.0 ± 46.2 ^c^
0	1	F	133	431.4 ± 108.9	262.5 ± 64.9	266.6 ± 65.0
0	1	M	93	541.3 ± 115.3 ^c^	290.4 ± 58.4 ^c^	309.0 ± 63.6 ^c^
1	0	F	286	427.3 ± 69.9	237.9 ± 37.0	263.5 ± 40.8
1	0	M	118	525.1 ± 75.1 ^c^	260.1 ± 37.4 ^c^	300.4 ± 42.7 ^c^
1	1	F	289	452.1 ± 97.2	253.1 ± 52.9	280.7 ± 59.1
1	1	M	230	627.7 ± 148.2 ^c^	306.6 ± 66.0 ^c^	356.2 ± 78.6 ^c^
2	0	F	215	451.6 ± 68.4	227.3 ± 32.1	279.2 ± 40.5
2	0	M	49	536.4 ± 72.4 ^c^	248.2 ± 32.8 ^c^	308.8 ± 40.9 ^c^
2	1	F	295	513.9 ± 124.4	257.3 ± 56.2	319.3 ± 74.8
2	1	M	244	652.9 ± 152.6 ^c^	291.7 ± 64.0 ^c^	372.9 ± 84.4 ^c^
Entire population			2305	499.4 ± 134.7	260.8 ± 56.4	299.0 ± 72.6

Abbreviations: BMI category 0—normal weight (BMI < 25 kg/m^2^), 1—overweight (BMI 25–29.9), 2—obesity (BMI ≥ 30); risk subgroup 0—CACS < 10, normal blood pressure, non-diabetic, 1—CACS ≥ 10 and/or high BP and/or t2DM and/or smoker and high lipids or structural abnormalities. F—female, M—male. Data from the first two rows (fonts in green) represent the reference group. Means and 1 standard deviation are present. Two-sided T-Student’s tests were used for comparisons between males and females, statistical significance: “c”—*p* < 0.001.

**Table 7 diagnostics-15-01502-t007:** Measured, expected, and standard THV concerning the h-BSA differences and sex, BMI class, and risk.

Sex	BMI	Risk	n	THVs (mL)	THVm (mL)	THVex (mL)	THV (m-ex. mL)	THV_h_-THV_BSA_ (mL)
F	0	0	285	406.0 ± 1.7	405.3 ± 6.3	410.9 ± 2.5	−5.6 ± 5.9	3.1 ± 1.0
F	0	1	133	402.1 ± 2.4	431.4 ± 9.3 ^b^	408.4 ± 3.6	23.0 ± 8.7 ^c^	4.1 ± 1.4
F	1	0	286	403.8 ± 1.7	427.3 ± 6.3	442.9 ± 2.4	−15.7 ± 5.9	25.7 ± 1.0
F	1	1	289	399.0 ± 1.7	452.1 ± 6.3 ^c^	438.0 ± 2.4	14.1 ± 5.9 ^c^	27.6 ± 1.1 ^a^
F	2	0	215	402.3 ± 1.9	451.6 ± 7.3	484.5 ± 2.8	−32.9 ± 6.8	51.9 ± 1.1
F	2	1	295	398.1 ± 1.6	513.9 ± 6.2 ^c^	483.3 ± 2.4	30.6 ± 5.8 ^c^	62.0 ± 1.0 ^c^
M	0	0	68	460.0 ± 3.4	506.9 ± 12.9	485.8 ± 5.0	21.1 ± 12.1	18.2 ± 2.0
M	0	1	93	463.2 ± 2.9	541.3 ± 11.1 ^a^	488.4 ± 4.3	52.9 ± 10.3 ^a^	18.6 ± 1.7
M	1	0	118	462.5 ± 2.6	525.1 ± 9.8	523.0 ± 3.8	2.2 ± 9.2	40.3 ± 1.5
M	1	1	230	466.7 ± 1.9	627.7 ± 7.0 ^c^	529.6 ± 2.7	98.2 ± 6.6 ^c^	49.6 ± 1.1 ^c^
M	2	0	49	457.3 ± 4.0	536.4 ± 15.2	551.6 ± 5.9	−15.3 ± 14.3	60.6 ± 2.4
M	2	1	244	462.5 ± 1.8	652.9 ± 6.8 ^c^	570.2 ± 2.7	82.8 ± 6.4 ^c^	81.2 ± 1.1 ^c^

Abbreviations: BMI category 0—normal weight (BMI < 25 kg/m^2^), 1—overweight (BMI 25–29.9), 2—obesity (BMI ≥ 30); risk subgroup 0—CACS < 10, normal blood pressure, non-diabetic, 1—CACS ≥ 10 and/or high BP and/or t2DM and/or smoker and high lipids or structural abnormalities. F—female, M—male. Data from the first two rows (bold fonts in green) represent the reference group. Statistics: means and 1 standard error of means are present. Two-sided Student’s *t*-test was used for comparisons between low- and high-risk males and females for each BMI class, with statistical significance: “a”—*p* < 0.05, “b”—*p* < 0.01, “c”—*p* < 0.001.

**Table 8 diagnostics-15-01502-t008:** Proportions of individual TVH above the expected value for sex, BMI class, and CV risk.

Sex	BMI Class	CV Risk	THV Expected	THV Abnormal	N
F	normal	low	274 (96.1)	11 (3.9)	285
		high	114 (85.7)	19 (14.3) ***	133
	total		388 (92.8)	30 (7.2)	418
	overweight	low	280 (91.2)	6 (8.8)	286
		high	247 (85.5)	42 (14.5) ***	289
	total		527 (91.7)	48 (8.4)	575
	obese	low	212 (98.6)	3 (1.4)	215
		high	228 (77.3)	67 (22.7) ***	295
	total		440 (86.3)	70 (13.7)	510
M	normal	low	62 (91.2)	6 (8.8)	68
		high	71 (76.3)	22 (23.7)	93
	total		133 (82.6)	28 (17.4) *	161
	overweight	low	109 (92.4)	9 (7.6)	118
		high	143 (62.2)	87 (37.8) ***	230
	total		252 (72.4)	96 (27.6)	348
	obese	low	48 (98.0)	1 (2.0)	49
		high	163 (66.8)	81 (33.2) ***	244
	total		211 (72.0)	82 (28.0)	293
Column total			1951 (84.6)	354 (15.4)	2305

Abbreviations: F—female, M—male, BMI—body mass index, CV—cardiovascular, THV—total heart volume [mL]. Statistics: chi-square test, * *p* < 0.05, *** *p* < 0.001.

**Table 9 diagnostics-15-01502-t009:** Total heart volume and the expected individual upper normal limit in the entire sample and subgroups.

	THV_UL_ ≤ 0	n	THV_UL_ > 0	n	% > 0	*p* < ¶
All	461 ± 93	1951	709 ± 136	354	15.4	0.001
F	428 ± 74	1335	635 ± 102	148	6.3	0.001
M	536 ± 87 ^#^	596	763 ± 133 ^#^	206	20.3 ***	0.001
Low	443 ± 82	985	582 ± 54	36	3.5	0.001
High	480 ± 100 ^#^	966	724 ± 135 ^#^	318	24.8 ***	0.001
Normal BP	449 ± 88	509	686 ± 100	68	11.8	0.001
High BP	466 ± 94 ^#^	1442	715 ± 143	286	16.6 **	0.001
T2DM Absent	460 ± 92	1569	709 ± 143	278	15.5	0.001
T2DM Present	468 ± 97	382	712 ± 112	76	21.5	0.001
<10 AU	453 ± 91	1007	717 ± 146	123	10.9	0.001
≥10 AU	469 ± 94	1065	707 ± 133	231	17.8 ***	0.001

Abbreviations: AU—Agatston units, BP—blood pressure, F—female, M—male, T2DM—type 2 diabetes mellitus, THV_UL_—total heart volume upper limit [ml]. Statistics: cChi-square test for “% > 0” ** *p* < 0.01, *** *p* < 0.001, ^#^ 0.1 > *p* > 0.05, ¶—2-sided Student’s t-test for THV_UL_ ≤ 0 vs. THV_UL_ > 0, 2-sided t-Student’s *t*-test for difference between within subgroups.

**Table 10 diagnostics-15-01502-t010:** Indexed THV with the expected upper normal limit in the entire sample and subgroups.

Group/Subgroup	Parameter	THV_UL_ ≤ 0	THV_UL_ > 0
All n = 2305		n = 2047	n = 258
	THV/h	281 ± 51	438 ± 70 ***
	THV/BSA	247 ± 55	369 ± 55 ***
	THV_h_-THV_BSA_	34 ± 25	69 ± 38 ***
F n = 1503		n = 1408	n = 95
	THV/h	268 ± 46	417 ± 63 ***
	THV/BSA	239 ± 38	358 ± 55 ***
	THV_h_-THV_BSA_	29 ± 24	59 ± 43 ***
M n = 802		n = 639	n = 163
	THV/h	311 ± 47	450 ± 71 ***
	THV/BSA	264 ± 36	375 ± 54 ***
	THV_h_-THV_BSA_	46 ± 25	75 ± 35 ***
Low risk n = 1021		n = 1011	n = 10
	THV/h	270 ± 46	377 ± 16 ***
	THV/BSA	242 ± 38	346 ± 12 ***
	THV_h_-THV_BSA_	28 ± 24	30 ± 21 ^NS^
High risk n = 1284		n = 1036	n = 248
	THV/h	293 ± 45	441 ± 70 ***
	THV/BSA	252 ± 39	370 ± 57 ***
	THV_h_-THV_BSA_	41 ± 27	71 ± 38 ***
Normal BP n = 577		n = 533	n = 44
	THV/h	273 ± 50	426 ± 44 ***
	THV/BSA	247 ± 40	370 ± 53 ***
	THV_h_-THV_BSA_	26 ± 23	57 ± 36 ***
High BP n = 1728		n = 1514	n = 214
	THV/h	284 ± 51	441 ± 74 ***
	THV/BSA	247 ± 39	370 ± 55 ***
	THV_h_-THV_BSA_	37 ± 26	71 ± 38 ***
T2DM Absent n = 1728		n = 1514	n = 214
	THV/h	284 ± 51	441 ± 74 ***
	THV/BSA	247 ± 39	369 ± 55 ***
	THV_h_-THV_BSA_	37 ± 26	72 ± 39 ***
T2DM Present n = 458		n = 399	n = 59
	THV/h	286 ± 53	435 ± 56 ***
	THV/BSA	242 ± 38	358 ± 39 ***
	THV_h_-THV_BSA_	44 ± 27	77 ± 37 ***
<10 AU n = 1009		n = 920	n = 89
	THV/h	276 ± 50	442 ± 71 ***
	THV/BSA	245 ± 39	372 ± 58 ***
	THV_h_-THV_BSA_	31 ± 26	71 ± 39 ***
≥10 AU n = 1296		n = 1127	n = 169
	THV/h	286 ± 51	436 ± 69 ***
	THV/BSA	249 ± 39	368 ± 53 ***
	THV_h_-THV_BSA_	37 ± 25	69 ± 38 ***

Abbreviations: AU—Agatston units, BP—blood pressure, F—female, M—male, T2DM—type 2 diabetes mellitus, THV_UL_—total heart volume upper limit [ml]. Statistics: 2-sided Student’s *t*-test for THV_UL_ ≤ 0 vs. THV_UL_ > 0 *** *p* < 0.001, NS—no significant difference.

## Data Availability

We have not asked the examined subjects to allow their data to be available, as the study started long before when such statements had not been required. A retrospective design of our study might make such an agreement to be possessed from examined subjects unlikely.

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
