# Peer review of "A Novel Concept of the “Standard Human” in the Assessment of Individual Total Heart Size: Lessons from Non-Contrast-Enhanced Cardiac CT Examinations"

_diagnostics, 2025, doi:10.3390/diagnostics15121502_

Round 1
Reviewer 1 Report
Comments and Suggestions for Authors
In the current paper, the authors assessed the impact of cardiovascular risk factors, including age, sex, hypertension, obesity, diabetes, CAC score, and anthropometric measures, including weight, height, BMI, and a novel indicator of h-BSA on THV. The findings were interesting using such a novel equation. Please calculate the SCORE scores of the patients and use this score rather than any risk factor separately. Using THV has less clinical value for the patients. Thus, more specific chamber quantification is necessary to predict future risks mentioned by the authors. Was there any correlation between THV/BSA vs. THV/h-BSA?
Author Response
Dear Reviewer 1,
Thank you for your careful reading of our manuscript and for your thoughtful and encouraging comments.
We fully appreciate your suggestion to recalculate cardiovascular risk using the SCORE system; however, several factors limit the feasibility and appropriateness of this approach in the context of our study.
First, the imaging modality employed—non-contrast cardiac computed tomography—was used primarily in asymptomatic individuals with cardiovascular risk factors to assess the presence and extent of calcified coronary atherosclerosis. In this population, coronary artery calcium (CAC) scoring has been shown to provide superior risk reclassification, particularly because traditional clinical risk scores, including SCORE, tend to overestimate risk by approximately 8 to 10 years. Thus, CAC scoring serves as a more refined and contemporary surrogate for global cardiovascular risk.
Second, and critically, our cohort also included symptomatic patients referred for cardiac CT, in whom non-contrast imaging often represents the initial—and occasionally final—stage of diagnostic evaluation. In these patients, contrast administration may not be possible due to extensive calcification or comorbid conditions. As a result, the non-contrast scan becomes the only available tool for assessing coronary pathology. We have now clarified this in the Methods section, explicitly stating that for such individuals, estimation of CAD probability—rather than SCORE-based risk stratification—is relevant. In fact, in our prior work, we demonstrated that CAC scoring retains strong prognostic power even in symptomatic populations.
Third, we must respectfully point out that accurate SCORE calculation requires objective clinical data, including blood pressure and cholesterol levels, which were not systematically collected in our imaging database. Risk factor assessment in this setting followed the standardized radiology department protocol, which relies on questionnaire-based reporting. While we recognize the inherent limitations of such data, this approach is consistent with the methodology adopted in numerous large-scale epidemiological and cross-sectional radiologic studies. Therefore, recalculating risk using SCORE would introduce significant bias and uncertainty, and would not align with either the design or scope of our study.
Clinically, it is also worth emphasizing that among hypertensive patients, those with poor treatment adherence, drug intolerance, or therapeutic inertia are known to have worse cardiovascular outcomes, regardless of nominal blood pressure values. Conversely, patients under well-individualized treatment plans with good adherence often fare better. Given that hypertension remains a leading contributor to heart failure, identifying subclinical cardiac remodeling—such as target organ damage detectable on non-contrast CT—offers an opportunity for early intervention, independent of the calculated SCORE risk. We believe this highlights the clinical utility of structural cardiac assessment in a broader context.
In this study, we selected total heart volume (THV) as a key parameter because it extrapolates the well-established cardiothoracic ratio and can be reliably measured on non-contrast CT. While atrial chambers can be clearly visualized, ventricular discrimination is not feasible with this imaging technique. However, recent advances in artificial intelligence now allow segmentation of myocardial tissue and blood pools, permitting estimation of left/right ventricular volumes and even left ventricular mass—without contrast or ECG gating. Although assessment of individual cardiac chambers is certainly desirable when technically possible, our findings indicate that global heart size serves as a robust proxy in studies focused on the impact of obesity and other systemic influences on cardiac structure. Predicting which chambers are most affected by obesity remains methodologically challenging, apart from the well-documented presence of epicardial fat and adipose tissue hypertrophy, which fell outside the scope of the present analysis.
Lastly, we thank you for your question regarding the relationship between THV/BSA and THV/h-BSA. If we have correctly understood your inquiry, our statistical analysis demonstrated no significant correlation between these two indices.
Once again, we are grateful for your time, your close reading, and your insightful suggestions, which have helped us to clarify and improve several key aspects of our manuscript.
With respectful appreciation,
Prof. Maciej Sosnowski MD, PhD, FESC, FEACVI
Reviewer 2 Report
Comments and Suggestions for Authors
I appriciate the opportunity to review such an interesting manuscript!
The authors are to be congratulated on a well-executed study and manuscript. The manuscript is well planned and written.
Authors hypothesized that the “standard human” body weight to height is expressed by the value of body surface area, which numerically equals height. Accordingly, when the difference between height (e.g. 1.73 [m]) and the BSA value (e.g. 91 1.73 [m2]) is equal to "zero" it is an indicator of the "standard human". Any deviation from this value determines the share of body mass regardless of height.
2,305 people who had NCE-CCT performed to determine the coronary artery calcium score. A statistical subject that has equal unitless values for height (i.e. 1.73) and BSA (i.e. 1.73). The arithmetic difference between these two numbers (height (i.e. 1.73) and BSA (i.e. 1.73), called h-BSA) above "zero", normally seen in the vast majority of individuals, indicates that body weight differs actually from that related solely to height for many reasons, like sex difference (at the same height, males weigh more), physical activity habits (more active people usually have a bigger heart), or fat tissue accumulation (overweight and obesity), among others.
The major strengths are the well-provided rather difficult reseach with a lot of respondents and parameters.
The major limitations are the lack of clinical data of a high risk group.
My suggestions are as follows:
-if its possible, it’d be better to divide people by cardio-vascular patology, e.g. coronary arthery disease, valve pathology, hypertension, atherosclerosis of carotid and/or femoral artery, cardiomyopathy
- Could authors calculate or sugest any coefficient for overweight/obese people?
The important asset of the “standard human” heart size is that the proposed equation, knowing the height and weight of individuals or the sample averages, allows for the re-494 calculation of normative data obtained by using various methods evaluating cardiac size and/or specific chamber size wherever these parameters were used for indexation. Our 496 results confirmed that indexation of the whole heart (as well as its constituents) for BSA introduces errors in overweight/obese people, that actually can be estimated and overcome by using the “standard human” concept we proposed. Since the presented empirical equations referred to a specific population sample, the average THV for a different sample should be adjusted accordingly
- is there cut-off point of THV >0 to predict complications or severity of acute illness in subgroups?
Author Response
Dear Reviewer 2,
Your warm words were both uplifting and motivating. We truly thank you for your kind appreciation.
Given the retrospective design of our study, the type of analysis you suggested was not feasible within our dataset, as numerous confounding factors could not be adequately controlled. Furthermore, some parameters were obtained through questionnaires containing self-reported qualitative data, which limits the precision and reliability of these measures. However, the key parameters for analysis were derived from objective measurements, providing a solid foundation for their usefulness in further analyses.
We appreciate your suggestion to stratify the cohort by specific cardiovascular pathologies such as coronary artery disease, valvular heart disease, hypertension, atherosclerosis of carotid and/or femoral arteries, and cardiomyopathies. While we recognize the value of such subgroup analyses, the present sample size of 2,305 individuals—although substantially larger than in many prior studies—is insufficient to support adequately powered subgroup analyses when further subdivided by BMI categories. Specifically, creating five diagnostic groups and three BMI strata would generate 15 subgroups, significantly reducing the number of patients per cell, especially considering the uneven distribution of conditions.
In our cohort, 75% of patients had hypertension, 20% diabetes, 63% coronary atherosclerosis, 53% were hyperlipidemic, and 21% were smokers. These prevalences highlight the frequent coexistence of multiple cardiovascular risk factors and diseases. For example, over 50% of patients with coronary artery disease also have systemic hypertension, and approximately 30% present concomitant peripheral atherosclerosis. Non-ischemic cardiomyopathies represent less than 10% of the cardiology cohort. Such overlap creates complex, non-exclusive subgroups that complicate statistical analysis and increase the risk of type II error due to small sample sizes within strata.
Statistical power calculations suggest that to enable meaningful and adequately powered stratified analyses across five major diagnostic subgroups and three BMI categories, the total cohort size should realistically approach or exceed 5,000 to 7,000 participants. However, the frequent coexistence and multiple possible combinations of cardiovascular diseases—such as coronary artery disease coexisting with hypertension and/or peripheral atherosclerosis—substantially increase the number of mutually exclusive subgroups. When each unique combination of pathologies is further stratified by three BMI categories, the total number of subgroups multiplies accordingly.
Importantly, the inclusion of less prevalent conditions such as non-ischemic cardiomyopathies, which constitute less than 10% of the cardiology cohort, further complicates stratification. Because these rare subgroups have low absolute numbers, a disproportionately larger overall sample size is required to achieve adequate power for statistical comparisons within these strata. Consequently, to maintain sufficient statistical power across all subgroups—including both common and rare pathologies—the total cohort size may need to exceed 10,000 participants.
Moreover, even if such a large cohort were assembled, it is unlikely that the results would significantly differ from those obtained in our current sample of 2,305 participants. This is because verification of the “standard individual” concept would most likely confirm its validity rather than challenge it. Hence, while larger studies could provide additional granularity, they would probably reinforce the fundamental findings presented here.
Regarding the second question, at this stage, we have derived a specific index for overweight or obese individuals, but we are still actively working to refine and overcome its weakness. The initial results have been presented at the Congress of the European Society in London in 2024.
Our current manuscript establishes the conceptual basis and highlights the challenges of the BSA-based and height-based indexing only. Verification of our robust and validated index remains an ongoing effort, which we plan to present in future work.
We believe this approach will contribute to more accurate and individualized cardiac size assessment, especially in populations with increased body mass. prognostic
As we did not plan to evaluate the prognostic value of our concept in relation to patient outcomes, attempting such an analysis in this heterogeneous population would be inherently invalid. If we were to conduct a separate study in asymptomatic individuals, in symptomatic patients evaluated specifically for CAD, or in an all-comers cohort, the prognostic significance of the proposed method for cardiac size normalization could be meaningfully assessed. At present, however, it is too early to conclude that the approach we propose is more clinically useful than current methods. The only advantages are, for now, grounded and logical, but remain to be confirmed by outcome-based evidence.
With respectful appreciation,
Prof. Maciej Sosnowski MD, PhD, FESC, FEACVI